# Dual PET-fMRI reveals a link between neuroinflammation, amyloid binding and compensatory task-related brain activity in Alzheimer's disease

Nádia Canário [1,2,3,5], Lília Jorge [1,2,4,5], Ricardo Martins [1,2], Isabel Santana[3] & Miguel Castelo-Branco [1,2,3 ✉]

The interplay among neuropathological mechanisms underlying Alzheimer's disease (AD), as neuroinflammation and amyloid-beta (Aβ), as well their impact on neuronal function remains elusive. A major gap in knowledge is the functional impact of neuroinflammation. The posterior cingulate cortex (PCC), as the most prominent site of amyloid pathology in AD, is a pivotal region to investigate the concomitant presence of pathophysiological mechanisms such as microglia activation, indexing neuroinflammation, and changes in task related activity. Here we used a dual PET approach to simultaneously study Aβ load and neuroinflammation (TSPO uptake marker), using $^{11}$C-PiB and $^{11}$C-PK11195 radiotracers, respectively and fMRI to study task related neural activation in an AD sample ($n = 19$) and matched controls ($n = 19$). Here we show significantly increased Aβ deposition, neuroinflammation and brain activity related to a visual object working memory task in this key region. Microglia activation was associated with increased brain activity specifically in patients, independently of amyloid binding, raising the possibility that abnormal brain activity might be restored in clinical trials aimed at reducing microglia activation.

[1] Coimbra Institute for Biomedical Imaging and Translational Research (CIBIT), University of Coimbra, Coimbra, Portugal. [2] Institute for Nuclear Sciences Applied to Health (ICNAS), University of Coimbra, Coimbra, Portugal. [3] Faculty of Medicine, University of Coimbra, Coimbra, Portugal. [4] Faculty of Sciences and Technology, Department of Physics, Coimbra, Portugal. [5] These authors contributed equally: Nádia Canário, Lília Jorge. ✉email: mcbranco@fmed.uc.pt

Pathophysiological and histopathological features in AD remain difficult to relate to causal mechanisms. This is the case concerning the role of abnormal Aβ aggregates and neuroinflammation in the pathogenesis of the disease[1]. The relationship between increased uptake of Aβ and microglia activation[2–4], remains uncertain as well as their association with regions underlying cognitive deficits[5,6]. One such region is the PCC, an area that seems to play a pivotal role in the pathophysiology of AD.

In fact, there is some evidence that puts the PCC as an excellent candidate for exploring the interplay between function and AD-related pathophysiological events. For instance, the PCC is a default mode network (DMN) region, involved in cognitive effort and episodic memory[7,8], being also a hotspot for AD-related neuropathological burden[3,9]. Previous studies have indeed associated PCC with both early abnormal Aβ deposition[7,10] and increased neuroinflammation[3,11], possibly due to its high synaptic activity, which increases the need for adequate autophagic response in order to maintain synaptic homeostasis[10]. Evidence of hypometabolism[10,12] along with altered resting state functional connectivity in this region has also been reported in AD patients[10,13,14]. Thus, the PCC appears to be a particularly vulnerable region for early neuronal dysfunction in this disease.

Nonetheless, a clear link between the pathophysiological features and neuronal function in specific brain regions is still controversial[2,4,15,16]. Previous studies exploring the functional impact of Aβ deposition and neuroinflammation in AD mainly resorted to resting-state functional connectivity, in which the link between the role of PCC disruption and cognitive function is hard to establish. We therefore attempted to overcome the limitations of former studies and directly investigated the relationship between both Aβ deposition and neuroinflammation with task related blood-oxygen-level dependent (BOLD) response in functionally defined PCC. For that, we conducted a multimodal neuroimaging study by combining positron emission tomography (PET) imaging with both radiotracers 11C-PiB and 11C-PK11195 to measure cerebral Aβ accumulation and neuroinflammation levels, respectively, and functional magnetic resonance imaging (fMRI) blood oxygenation level dependent (BOLD) response, in a demanding task that implies recognition skills of complex visual features, prompting the PCC activation.

As the main result, we found that microglia activation in the PCC was associated with increased task-related brain activity in AD and was independent of amyloid accumulation.

## Results

**Demographic data results.** We were able to collect imaging data from 19 AD patients (all 11C-PiB positive) and 19 cognitively normal controls (16 11C-PiB negative and 3 11C-PiB positive). The control group was composed of 19 age (t (36) = 0.024, $p > 0.980$), gender ($p > 0.999$) and education matched (U = 127, $p > 0.111$) participants. Clinical and demographical information are displayed in Table 1. As expected, AD patients evidenced

significant lower MoCA scores relative to controls. Concerning BLAD (Battery of Lisbon for Evaluation of Dementias), AD patients showed the following ratios of impairment: (Memory, 16/17; Executive, 12/17; Language, 7/17; Constructive, 4/17; Calculation, 3/17). CSF data collected from 17 AD patients revealed abnormal values of Aβ, Tau and $p$Tau (Table 2), being consistent with 11C-PiB positivity.

**Behavioural results.** Behavioural results revealed a main effect for stimuli for both (AD: $\chi^2(5) = 35.06$, $p = 0.000001$/Controls: $\chi^2(5) = 36.15$, $p = 8.8674E-7$), showing that the scrambled stimulus was the most difficult stimuli to process (Table 3). Between group differences were found for the scrambled stimuli, with the AD group showing a lower d'prime compared to controls groups (AD: mean (−0.08), s.d (1.63), N = 19; controls: mean (1.95), s.d (1.46), N = 19 ($p_{corrected} = 0.001332$)).

**Neuroimaging results.** All brain regions identified by the functional task are depicted in Fig. 1. At the ROI level, we found in our PCC region of interest, a multivariate effect for the three brain imaging measures explored ($X^2$ (3) = 23.236, $p = 0.000036$). Post hoc analysis revealed a significant higher Aβ deposition (AD: mean (1.918), s.d (0.276), N = 19; controls: mean (1.227), s.d (0.311), N = 19; $X^2KW$ (1) = 21.143, $p = 0.000004$), neuroinflammation (AD: mean (0.188), s.d (0.103), N = 18; controls: mean (0.101), s.d (0.083), N = 19; ($X^2KW$ (1) = 5.763, $p = 0.016370$) and brain activity (beta values) (AD: mean (−0.236), s.d (0.260), N = 19; controls: mean (0.599), s.d (0.275), N = 19; ($X^2KW$ (1) = 12.582, $p = 0.000389$) in the PCC for the AD group (see Fig. 2 for Aβ deposition and neuroinflammation). Group differences in 11C-PK11195 binding potential ($BP_{ND}$) for the remaining ROIs found in whole-brain General Linear Model (GLM) procedure are depicted in Supplementary Table 1. Source data underlying Fig. 2c, f are available in Supplementary Data 1, 2, respectively.

Remarkably we found a significant positive correlation between 11C-PK11195 binding potential ($BP_{ND}$) and the beta values in the AD group ($r = 0.586$, $p = 0.013476$) whereas no significant association was found in the control group ($r = 0.327$, $p = 0.185308$) (see Fig. 3). No other relevant correlation was found between the 11C-PK11195 binding potential ($BP_{ND}$) and the beta values in the remaining ROIs. The 11C-PiB standardized uptake value ratio (SUVR) did not associate with brain activity (AD: $r = 0.004$, $p = 0.986140$/ controls: $r = 0.133$, $p = 0.599614$) and 11C-PK11195 $BP_{ND}$ was also not associated with 11C-PiB SUVR (AD: $r = 0.196$, $p = 0.450431$/controls: $r = −0.181$, $p = 0.472908$). Finally, also no significant correlations were found among global CSF driven Tau values and PCC neuroinflammation, brain activity or Aβ.

## Discussion

In the present study we sought to investigate the link between AD pathophysiological hallmarks and task related brain activity in the

**Table 1 Demographics and brief clinical variables.**

| | Alzheimer's disease (n = 19) | Controls (n = 19) | p value |
|---|---|---|---|
| Age (mean ± sd) | 66.11 ± 7.02 | 66.05 ± 6.77 | >0.980 |
| Education (mean ± sd) | 8.95 ± 5.83 | 10.53 ± 5.31 | >0.111 |
| Gender (m: f) | 10:9 | 10: 9 | >0.999 |
| MoCA (mean ± sd) | 14.26 ± 4.31 | 24.94 ± 3.62 | <0.001 |
| MMSE (mean ± sd) | 23.1 ± 2.97 | – | |
| ApoE-ε4 (%) | 95% | – | |

Demographics and performance on MoCA for both AD and controls. For the AD group the MMSE scores are also shown and ApoE which is expressed as percentage of ε4 carriers.

**Table 2 CFS's parameters of AD patients.**

| | $A\beta_{1-42*}$ (n = 17) | $A\beta_{42}/A\beta_{40}$* (n = 14) | Tau* (n = 17) | pTau* (n = 17) | Tau/ $A\beta_{42}$* (n = 17) | $A\beta_{42}$/pTau* (n = 17) |
|---|---|---|---|---|---|---|
| Mean | 510.94 | 0.057 | 445.64 | 63.15 | 0.98 | 9.32 |
| SD | 215.00 | 0.021 | 246.80 | 24.36 | 0.52 | 6.28 |

Mean and standard deviation for the CSF's parameters. Note: *Normal values: $A\beta_{42} > 580$ pg/mL; $A\beta_{42}/A\beta_{40} > 0.068$; Tau < 250 pg/mL; $p$Tau < 37 pg/mL; Tau/$A\beta_{42} < 0.40$; $A\beta_{42}/p$Tau > 15.8.

**Table 3 Descriptive statistics for the behavioural task.**

| | Stimuli | Dprime (mean ± s.d) | Reaction Time (mean ± sd) |
|---|---|---|---|
| Alzheimer's Disease | Bodies | 1.05 ± 1.02 | 679.49 ± 392.22 |
| | Places | 2.13 ± 2.42 | 613.50 ± 191.36 |
| | Verbal | 1.83 ± 1.84 | 699.92 ± 175.82 |
| | Scrambled | −0.08 ± 1.63 | 650.96 ± 226.08 |
| | Objects | 1.58 ± 1.88 | 642.68 ± 188.06 |
| | Faces | 1.71 ± 1.17 | 720.47 ± 247.35 |
| Controls | Bodies | 3.45 ± 2.06 | 681.93 ± 102.40 |
| | Places | 4.03 ± 2.28 | 668.16 ± 90.98 |
| | Verbal | 4.23 ± 2.45 | 699.27 ± 91.00 |
| | Scrambled | 1.95 ± 1.46 | 643.08 ± 95.85 |
| | Objects | 3.41 ± 1.94 | 663.34 ± 84.79 |
| | Faces | 4.36 ± 2.44 | 691.95 ± 84.49 |

Mean and standard deviation (s.d) for both the dprime and reaction times obtained for each stimulus categories. Table depicts both AD and controls participants.

PCC, a cortical region particularly prone to neuropathological burden even in mild stages of AD. As the main result, we found regional task related brain activity associated with neuroinflammation in contrast to Aβ load. The lack of direct coupling between Aβ and our functional measures is in line with previous work where no associations with cognitive test scores[17–20] or with cerebral glucose hypometabolism[21] were found. Previous studies have suggested that regional neuroinflammation might accompany hypometabolism and brain shrinkage[22,23] being indeed recently related to cortical atrophy[4], impairments in brain connectivity[16] as well as to glucose hypometabolism[24]. Moreover, a linkage with cognitive performance, as measured by cognitive tests, has also been proposed[11,18]. As such, while it is possible that neuroinflammation could be the direct mediator of functional disturbances, the converse possibility might also hold true, e.g. functional disturbances might instead induce neuroinflammation[10].

Accordingly, our results provide a tantalizing association between neuroinflammation and brain function, as measured for the first time, to the best of our knowledge, using BOLD activation, in a target region of known AD pathophysiology which shows a notable overlap between higher levels of Aβ and neuroinflammation[3] as well as altered functional brain response[10,13,14] at a mild AD stage.

Despite a consensus regarding neuroinflammation as an important pathologic event in AD[16,25], its interplay with Aβ deposition and brain function continues to be highly debated. This striking association between activated microglia and cognition found in our study, in the presence of high amyloid load, that was by itself not directly related with brain activity, might suggest a dissociation between the two neuropathological molecular mechanisms: increased levels of activated microglia in the presence of elevated levels of Aβ, are independently related to cognitive functioning and neural activation.

Thus, the effects of neuroinflammation in the function of PCC might not be mediated by Aβ deposition, and as such, our results are not consistent with the notion that cortical Aβ accumulation is a causative agent of functional impairment[26] in mild AD, even though it may play a crucial role at asymptomatic stage[21,27]. Our findings

suggest instead that neuroinflammation has an independent active role in cognitive impairment or is instead a pathologic product of brain dysfunction at a symptomatic AD stage.

We acknowledge however that some previous studies have suggested that regional associations between neuroinflammation and Aβ burden may occur[11,28,29]. Nevertheless, we delineated regions selectively affected by neuropathological mechanisms of AD through an fMRI task, which could explain the different results observed in the present study. Moreover, neuroinflammation has been found closer to Aβ in early MCI, whereas in established AD it has been preferentially linked to Tau tangle burden[30,31]. Nevertheless, we found no significant associations between global CSF values related to Tau and neuroinflammation in PCC, possibly because CSF measures are related to global brain values and not specifically to Tau in PCC, as analysed in our approach.

Despite the interesting findings of this study, some considerations should however be made. For instance, the potential influence of the rs6971 SNP on the binding affinity of the [11]C-PK11195 was not addressed. In fact, this PET tracer had reduced affinity to the mitochondrial TSPO compared to second-generation tracers[32,33]. Moreover, the relatively modest sample size of our groups should be considered as well.

Future studies should further address the local pathophysiological role of neuroinflammation in modifying brain activity and behaviour.

## Methods

**Participants.** A total of 19 patients in the mild phase of AD (Clinical Dementia Rating = 1) and 19 controls, recruited from the community with no evidence of cognitive impairment or other relevant pathologies, were enrolled in this study. Patients were recruited at the Neurology department of the Centro Hospitalar e Universitário de Coimbra (CHUC), where AD diagnosis was made by two experienced neurologists at the Memory Clinic of the Neurology department of CHUC. The clinical diagnosis was supported by biological biomarkers (cerebrospinal fluid - CSF and/or [11]C-PiB PET SUVR).

Regarding [11]C-PiB PET SUVR, its positivity was determined by an experienced nuclear medicine medical doctor relying on the regional visual analysis of the images, with emphasis to frontal, parietal/precuneus, temporal, anterior and posterior cingulate, basal ganglia and occipital cortices (a homemade machine learning algorithm using the SUVR values as features was also used to aid the diagnosis).

The CSF were collected in 17 patients from the AD group. In short, after 8 h of resting CSF samples were collected, in sterile polypropylene tubes, instantly centrifuged at 1800 g for 10 min at 4 °C, aliquoted into polypropylene tubes and stowed at −80 °C until analysis[34]. We have used the commercially available sandwich ELISAs (Innotest, Innogenetics/Fujirebio, Ghent, Belgium), to measure in duplicate CSF $A\beta_{1-42}$, $A\beta_{1-40}$, Tau, and $p$Tau[35]. Thereafter, the quality control scheme of the Alzheimer's Association Quality Control Program for CSF Biomarkers[36] was followed to guarantee External quality control of the measurements. Finally, the cut-off values applied in this work, were: 580 pg/mL for $A\beta_{1-42}$, 0.068 for $A\beta_{42}/A\beta_{40}$, 250 pg/mL for Tau, and 37 pg/mL for $p$Tau (Table 2).

In order for the imaging data to be collected all participants underwent one fMRI session and two PET sessions - with both [11]C-PiB and [11]C-PK11195 radiotracers, in 3 distinct visits to ICNAS (Institute for Nuclear Sciences Applied to Health). One AD patient did not perform the [11]C-PK11195 session.

The methods were performed in accordance with the tenets of the Declaration of Helsinki and approved by the Ethics Committee of the University of Coimbra. All participants provided written informed consent to take part in the study.

**fMRI stimuli and procedures.** All stimuli presented in this study were designed in order to map regions of the ventral visual stream, which were validated in a previous project performed on visual recognition in normal and pathological ageing[37]. Thus, a total of 6 types of visual stimuli were presented in this study - images of faces, bodies, objects, places, verbal material and scrambled images,

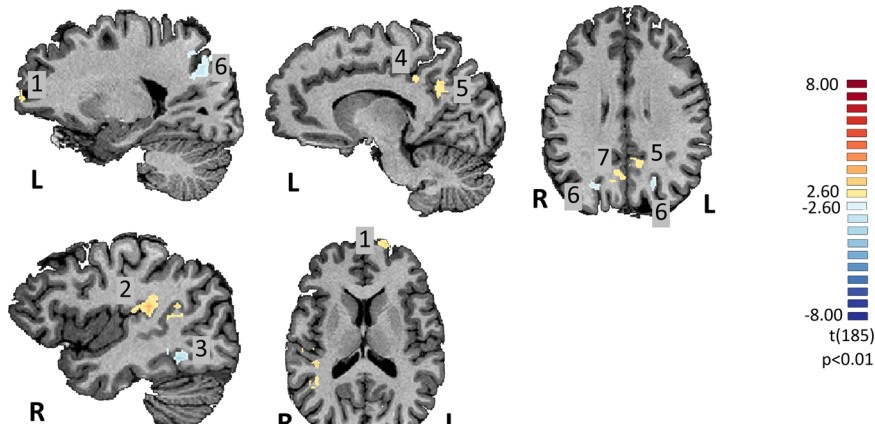

**Fig. 1 Whole-brain BOLD activation map.** Statistical BOLD map depicting all ROIs obtained from the contrast Scrambled AD > Scrambled Controls. Legend: 1 = left prefrontal cortex; 2 = right posterior insula; 3 = right fusiform gyrus; 4 = left ventral posterior cingulate gyrus; 5 = left posterior cingulate cortex; 6 = right/left parietal region (BA7) (deactivation ROI); 7 = right parietal region (BA7). ROI 5 depicts the PCC (posterior cingulate cortex), as it is the region under study.

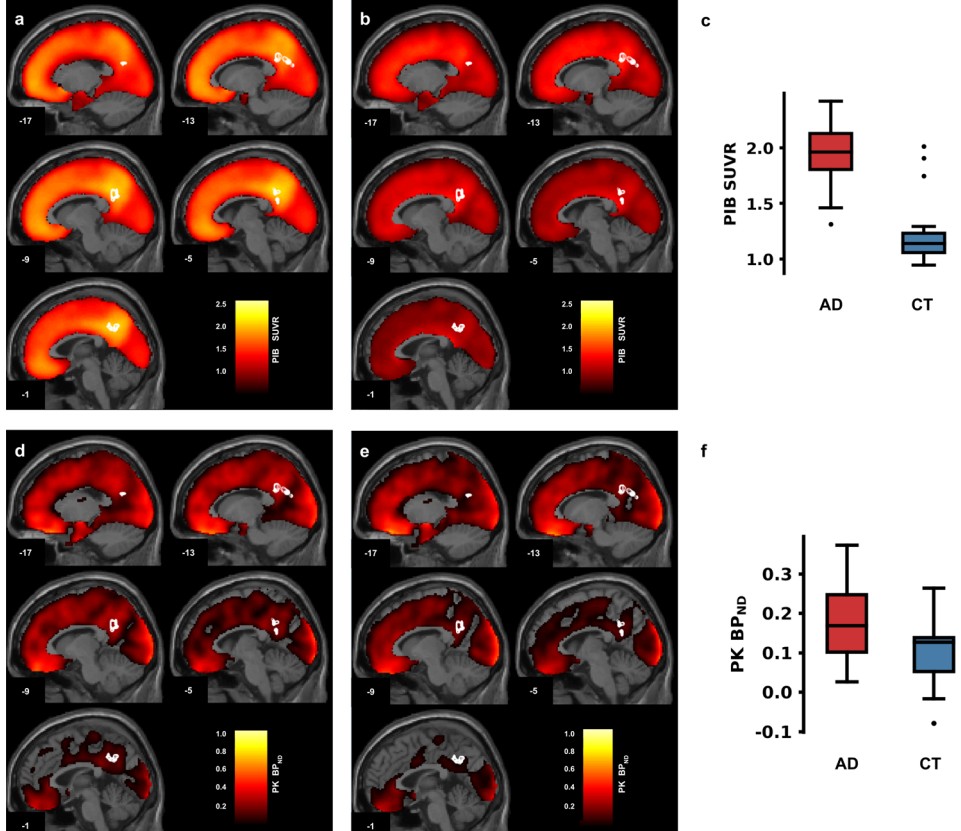

**Fig. 2 Aβ deposition and extension of neuroinflammation maps across the cortex.** The PCC cluster borders are highlighted in white. **a** Mean [11]C-PiB PET SUVR map of the AD group. **b** Mean [11]C-PiB PET SUVR map of the control group. **c** [11]C-PiB PET SUVR at PCC cluster in AD and control groups. **d** Mean [11]C-PK11195 PET $BP_{ND}$ map of the AD group. **e** Mean [11]C-PK11195 PET $BP_{ND}$ map of the control group. **f** [11]C-PK11195 PET $BP_{ND}$ at PCC cluster in AD and control groups. The subfigures (**c**) and (**f**) display the standard representation of a boxplot (median, first and third quartile, non-outlier maximum and minimum, and the outliers - values that are more than 1.5 x IQR away from the top or bottom of the box, IQR: interquartile range).

performing a total of six different categories of visual stimuli[37]. All images were grey-scale stimuli and were composed by three different subcategories - faces category was composed by faces of young, middle age and old persons; the body category showed body images, images of hands and feet and also body shape silhouettes; the objects category comprised tool, cars and chairs; the places category showed landscapes, buildings and skylines; the verbal stimuli comprised word, pseudowords and nonwords; and lastly scrambled category consisted of images with no semantic meaning. Stimuli were taken from different sources: face stimuli

were taken from the FACES database[38], faceless body images were taken from Bochum Emotional Stimulus Set database[39], whereas images of hands and feet were selected from publicly available online images body shape silhouettes were created using a customized code in MATLAB R2014a (MathWorks, Natick, USA); images from the places' category were part of the database of the computational visual cognition laboratory[40] (http://cvcl.mit.edu/database.htm); stimuli from the objects' category were taken from public online images; whereas verbal material was provided as a courtesy from the database of Universidade Católica Portuguesa, created

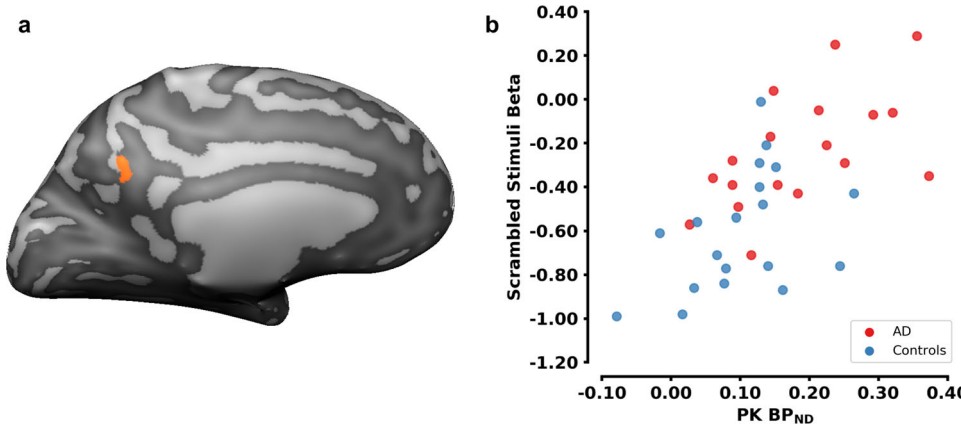

**Fig. 3 The interplay between neuroinflammation with the BOLD response at a task-driven functionally defined PCC cluster. a** PCC cluster (left hemisphere; center of mass in TAL space: −9.70, −55.44, 26.77; center of mass in MNI space: −9.16, −54.27, 31.2) extracted from the functional contrast [scrambled$_{AD}$ > scrambled$_{controls}$]. **b** Relationship between the extension of neuroinflamation at the PCC cluster, quantified by the $^{11}$C-PK11195 BP$_{ND}$, and the scrambled stimulus betas on that region. Higher betas express higher neuronal activation.

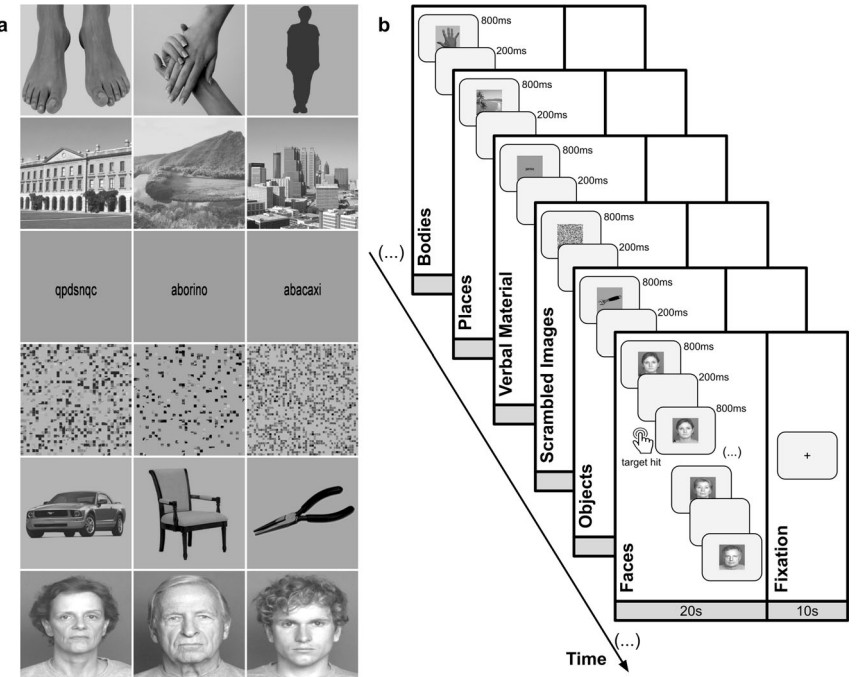

**Fig. 4 Task-driven functional mapping of PCC. a** fMRI task design, a total of 6 visual stimulus types were presented. Each run comprised 18 pseudo-randomized blocks with 3 blocks per category of objects. Each block was composed of 20 images, each one presented for ~800 ms followed by a ~200 ms of interstimulus interval which represents an individual duration of 20 s per block. Each block was separated by 10-s fixation. Subjects performed a 1-back task, being instructed to press a button every time the image being presented was the same that had been presented immediately before. **b** Sample images from different categories used on the fMRI task. Faces stimuli were taken from the FACES database; Bodies stimuli were selected from Bochum Emotional Stimulus Set database and Place stimuli were available on the Computational Visual Cognition Laboratory. **c** Behavioural metrics, dprime and reaction times (RT), obtained for both AD and controls in the functional task.

for previous research. Importantly, for the scrambled images we used a custom written algorithm in MATLAB, which divides each intact image into a grid of size 50 × 50 and 40 × 40, with the tiles being randomly shuffled, and filling 0.22° and 0.28° of the visual field, respectively. All stimuli were equalized for luminance and contrast with SHINE toolbox[41]. Examples of the visual categories used in this experiment are depicted in Fig. 4a.

The fMRI experiment was composed of two functional block-design runs. Each run comprised 18 pseudo-randomized blocks with 3 blocks per category of objects, meaning that scrambled images were presented in a total of three different blocks. Each block was composed of 20 images, each one presented for ~800 ms followed by a ~200 ms of interstimulus interval which represents an individual duration of 20 s per block. Each block was separated by 10-s fixation with uniform grey-scale image baseline interval, representing the baseline condition (Fig. 4b). Subjects performed a 1-back task during the scan session, meaning that they were instructed to press a button every time the image being presented was the same that had been presented immediately before. Each block always had four repetitions of images, that is four possible targets to which participants had to respond to, making a total of 12 chances for hits per stimulus category in each run. All subjects performed a brief training session (~60 s) before scan session allowing them to be familiarised with the stimulus type and task demands. All stimuli presented during the functional runs were delivered by a computer onto a LCD screen with 1920*1080 resolution at the head of the scanner with an angled mirror positioned on the head-coil, and were presented using Presentation 17.1 software (Neurobehavioral systems). The images' size used to build the stimuli was 544 × 544 pixels and subtended approximately 11° x 11° of visual field. Behavioural responses were collected during acquisition via a response box and stored in a log file.

**fMRI data acquisition**. In the first visit at Institute of Nuclear Sciences Applied to Health (ICNAS), University of Coimbra, MRI data were acquired in a 3 Tesla Siemens Magneton Trio scanner with a 12-channel head matrix coil. Each session started with a T1-weighted 3D anatomical MPRAGE (rapid gradient-echo) sequence, with a voxel resolution of $1.0 \times 1.0 \times 1.0$ mm, repetition time (TR) of 2530 ms, echo time (TE) of 3.42 ms, inversion time (TI) of 1100 ms and a field of view (FOV) of $256 \times 256$ mm. Each MPRAGE sequence comprised 176 slices, a flip angle of 7° and with interleaved series. T2*-weighted 2D echo-planar images composed the functional runs and had the following parameters: voxel size of $2.5 \times 2.5 \times 3$ mm, TR of 2000 ms, TE of 30 ms, FOV of $256 \times 256$ mm, matrix size of $102 \times 102$ and a flip angle of 90°. Each functional localizer sequence comprised 31 slices and had 276 volumes.

**fMRI data preprocessing**. All anatomical data in native space were corrected for the inhomogeneity of signal intensity, re-oriented into the AC-PC plane and further transformed to the Talairach reference system (TAL). In turn, functional data were corrected for differences in time for each slice, were applied a filter to remove low-frequency drifts and adjusted for mean intensity. All volumes were also corrected for motion. The pre-processed fMRI data were then initially coregistered with the anatomical native space followed by a transformation to Talairach space using the anatomical data previously converted to this reference system. The pre-processing of functional data along with further fMRI statistical procedures (see statistical analysis below) were computed in Brainvoyager QX 2.8.2 (BrainInnovation, Maastricht, the Netherlands).

**PET data acquisition**. All participants underwent $^{11}$C-PiB PET (second visit) and $^{11}$C-PK11195 PET scans (third visit) at the Institute of Nuclear Sciences Applied to Health (ICNAS), University of Coimbra[3] as summarized in Supplementary Fig. 1. Prior to radioligand injection ($^{11}$C-PiB PET or $^{11}$C-PK11195 PET), each participant had a low-dose brain CT scan for attenuation correction. The $^{11}$C-PiB PET or $^{11}$C-PK11195 PET image acquisition sessions started immediately after the intravenous bolus injection of approximately 555 MBq of $^{11}$C-PiB or 370 MBq of $^{11}$C-PK11195.

The dynamic $^{11}$C-PiB PET image consists of 24 frames (total duration of 90 min: 37 frames: $4 \times 15\,s + 8 \times 30\,s + 9 \times 60\,s + 2 \times 180\,s + 14 \times 300\,s$) and the dynamic $^{11}$C-PK11195 image of 22 frames (total duration of 60 min: $4 \times 30\,s + 4 \times 60\,s + 4 \times 120\,s + 4 \times 240\,s + 6 \times 300\,s$).

The whole-brain CT and PET data were acquired using a Philips Gemini GXL PET/CT scanner (Philips Medical Systems, Best, the Netherlands). The patient's head was restrained with a soft elastic tape to minimize head movement. The PET images were reconstructed to a $128 \times 128 \times 90$ matrix, with 2 mm isotropic voxel dimension, using the LOR RAMLA algorithm (Philips PET/CT Gemini GXL) with attenuation and scatter correction.

**PET data preprocessing and quantitative analysis**. The PET image preprocessing and quantitative analysis pipeline was analogous for both types of PET data acquisition sessions, $^{11}$C-PK11195 or $^{11}$C-PiB[3]. For each session, a rigid transformation was estimated, using 3D Slicer 4.8.1 software (http://www.slicer.org)[42], between the $^{11}$C-PiB PET sum image space or $^{11}$C-PK11195 PET sum image space and the T1 anatomical MRI space of each participant. SPM12 DARTEL algorithm[43] was used to spatially normalize the MRI scans to the Montreal Neurological Institute (MNI) space.

The voxel-level quantitative analysis of $^{11}$C-PiB PET images and $^{11}$C-PK11195 PET images was implemented in the MNI space using a software validated in previous works[3,44,45]. The subject specific $^{11}$C-PiB standard uptake value ratio (SUVR) map was calculated by summing voxel-level signal from 40 to 70 min post-injection, and dividing by the mean signal from the individual's reference region, the cerebellar grey matter[45,46].

The $^{11}$C-PK11195 BP$_{ND}$ maps were generated using the MRTM2 (Multilinear Reference Tissue Model 2)[47]. The time-activity curve of a reference region was determined by the algorithm SVCA4 (Supervised Cluster Analysis with 4 classes: grey matter without specific binding, white matter, blood, grey matter with specific binding)[48] which selected a group of grey matter voxels showing a time-activity curve representing the kinetic activity of normal grey matter without $^{11}$C-PK11195 specific binding.

The regions of interest (ROI) extracted from the fMRI data analysis (TAL space) were converted to the MNI space using GingerALE software v. 3.0.2 (http://brainmap.org/ale/). For each participant, the ROI mean values of $^{11}$C-PK11195 BP$_{ND}$ and $^{11}$C-PiB SUVR were extracted from the corresponding maps using 3D Slicer software.

Please, see flowchart on processing pipeline in Supplementary Fig. 1.

**Statistics and reproducibility**. In order to test which stimuli imposed more cognitive load, we resorted to d'prime [d' = $Z_{Hit} - Z_{FA}$] as a sensitivity measure and performed a Friedman test for both groups with subsequent post-hoc testing. Group comparisons were made with the Mann-Whitney test using Bonferroni correction. This procedure was particularly important since we aimed to map regions implicated in cognitive effort, including the PCC. As such, we further mapped brain activation areas whose functional contrast was defined according to the stimuli that imposed a higher cognitive demand (the scrambled object versions) during the 1-back task by performing a multisubject random-effect general linear model (RFX-GLM), with the contrast [scrambled$_{AD}$ > scrambled$_{Controls}$]. Statistical t maps were obtained with a threshold $p < 0.01$ corrected for multiple comparisons at a cluster level. In addition, subject-specific beta values for scrambled stimuli were extracted for the PCC cluster region. Non-parametric one-way MANOVA was performed in order to test for group differences regarding the three brain measures under analysis - $^{11}$C-PiB SUVR, $^{11}$C-PK11195 BP$_{ND}$ and beta values for the BOLD activity data. Beta values, or beta coefficients, are estimated effects of a particular condition that enters the General Linear Model (GLM) procedure. Thus, higher betas simply mean higher neuronal activation.

We then computed partial correlation tests, controlling for age, between each one of the molecular data - $^{11}$C-PiB SUVR and $^{11}$C-PK11195 BP$_{ND}$ - with the beta values of brain activation for the scrambled stimuli. Partial correlations between only $^{11}$C-PiB SUVR and $^{11}$C-PK11195 BP$_{ND}$ data were also performed. Further correlation analyses between the neuropathological markers and beta values for the other remaining ROIs were also performed. We have in addition computed correlation tests among CSF related Tau values and PCC neuroinflammation and Beta values.

The non-parametric one-way MANOVA and correlation analysis were conducted on IBM SPSS statistics software (version 27).

**Reporting summary**. Further information on research design is available in the Nature Research Reporting Summary linked to this article.

## Data availability

The source data underlying Fig. 2c, f are available in Supplementary Data 1 and Supplementary Data 2, respectively. The remaining datasets analysed during the current study are available from the corresponding author on reasonable request.

## Code availability

Preexisting/public code was used for the study[45].

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

## Acknowledgements

The authors would like to thank all the volunteers that participated on the study as well as to the ICNAS MRI and PET technicians. This work was supported by Neuroscience Mantero Belard Prize 2015 – Santa Casa da Misericórdia (MB-1049-2015) and the Foundation for Science and Technology, Portugal (UID/04950 B&P /2020, PAC-MEDPERSYST, POCI-01-0145-FEDER-016428, BIGDATIMAGE, CENTRO-01-0145-FEDER-000016, Centro 2020 FEDER, COMPETE, PTDC/PSI-GER/30852/2017, COV-DATA - DSAIPA/DS/0041/2020, PTDC/PSI-GER/1326/2020).

## Author contributions

M.C.B. conceived and designed the experiments; I.S., L.J., N.C. acquired data; N.C., L.J. and R.M. analysed the data; N.C., L.J. and M.C.B. wrote the article; R.M. and N.C. prepared the images; M.C.B. contributed significantly to discussion and helped revise the manuscript. N.C. and L.J. contributed equally to the work.

## Competing interests

The authors declare no competing interests.
