## [Peer Review File · Communications Biology]

Reviewers' comments:

Reviewer #1 (Remarks to the Author):

This is a brief report of 19 persons with A+ cognitive impairment (presumed due to AD) versus 19 control cognitively unimpaired people (not elevated amyloid) and the relationship of increase in BOLD signal in posterior cingulate in relation to a) amyloid PET levels (11C PIB) and b) a marker of microglial activation (PK11195). The authors find that there was a strong correlation between BOLD signal in the task in fMRI and microglial activation PET tracer, and no relation to amyloid. While the authors make an (unjustified) therapeutic speculation about the direction of causality of the association, the association is nonetheless of interest. The authors consider only that increased microglial activation was causal, its possible and indeed plausible that the reverse is true, namely that the increased activation is causing the increased microglial activation. See a discussion in Knopman et al, "Alzheimer Disease Primer" Nature Reviews 2021 on connectivity and its relationship to AD pathology.

Thus, the observations are quite interesting but the interpretation is questionable, and needs to be balanced by the alternative perspective.

From a presentation perspective, this report was very difficult for a reader to grasp because of the lack of detail and explanation of key issues in the very terse Results section. The main one was that "beta values" was not defined in text, and it took a good deal of scrambling by a reader to appreciate that that term referred to the activations of fMR – there is a mention of what "higher betas" refers to in the legend of Fig 2 and they were referred to again in the Statistical analysis, but there should have been a simple statement in text in Results that clarified the point. Second there should be a Table with demographics and basic disease descriptors of the cases and the comparison group. Third, I would have liked to see a Figure depicting PIB vs PK11195, in order to understand the relationship of cases to controls

One other major deficiency was that the relationship to other regions was not described: there is no presentation of the specificity of the findings for post cingulate. Was the BOLD signal – PK11195 signal association present elsewhere? Were there associations with PIB and BOLD elsewhere? These could be included as supplemental files but the reader needs to know.

Reviewer #2 (Remarks to the Author):

The current study investigated the association between task-fMRI activation, microglia PET and amyloid PET across AD patients and cognitively normal controls. Results showed that fMRI activity and microglia PET were positively associated in the PCC.

Overall is the topic of interest. But several concerns compromise the conclusiveness of the results.

- The rationale for focusing only on the group contrast for scrambled stimuli in all subsequent analyses and discarding the remainder of the fMRI data is unclear. Also it seems that the voxel-wise group comparison of fMRI activation between AD vs controls just happened to be significant in the PCC and was therefore used for further analysis, whereas the introduction renders the impression, that the PCC was selected a priori as the region of interest. Please clarify.
- Were there no brain regions of reduced brain activations in AD compared to controls? This seems surprising. Could the authors provide an explanation for this finding and was that consistent for other stimuli as well?
- Why was a multivariate group comparison done rather than analyzing each modality separately? The group differences should be mapped for each modality, so that it is clear in which brain regions increases/decreases in AD were observed.
- In Figure 3 it is hard to discern whether the controls showed abnormally high regional PiB as well. Again, the group comparison would be informative. Also, in the methods it is stated that CSF and PIB value were used in order to define abnormal Abeta values for the AD group, but the criteria were not mentioned.
- Were all cross-modality correlational analyses done only for the PCC ROI values?
- The lack of a correlation between PiB PET and microglia PET is surprising and contradicts previous reports such as in Hamelin et al. Brain 2016. Please discuss.

- On p. 4 (Discussion) the authors refer an association between "activated microglia and cognition", but an association with cognitive performance was not reported. In fact, that could have been an interesting addition.
- Please discuss caveats of the current study
- The potential influence of the rs6971 SNP on binding affinity of the microglia PET tracer is not reported

Minor:

In order to follow the Results section, basic methodological information should be provided to understand what the results refer to. For example, the authors start the Results section by mentioning that the scrambled stimuli were the most difficult to process, but no information on the fMRI paradigm was provided, which renders this result difficult to interpret.

Reviewer #3 (Remarks to the Author):

The multimodal imaging study by Canario et al is well designed and of high-interest given the importance of microgliosis in the development of Alzheimer disease. The author described the association between neuroinflammation (TSPO PET), amyloid (PIB PET) and compensatory task related brain activity (fMRI) in 19 AD and 19 control subjects. The author found that microglia activation in the posterior cingulate cortex was associated with increased brain activity in AD patients and independent of amyloid accumulation. More details in the method and results need to be provided to enable understanding of the finding.

1. Method: Lack of information regarding patient demographics, e.g. age, APOE, gender, MMSE score, disease duration, education. The author only mentioned that the participants are age, gender and education matched without providing detailed information. Neuroinflammation is dynamic and act differently at different disease stage. Thus these information are essential.
2. Method: the author mentioned that the diagnosis was supported by CSF assay. Please specify which CSF assay, and included the results. Is there any correlation between the CSF results and the imaging readouts.
3. Results: The PK11195 results are not presented. Please add a figure of PK11195 distribution and comparison of BPnd values between groups.
4. Please add the cognitive assessment data (if available) which is important for interpretation of the results
5. Please provide quantification and comparison of the regional SUVR for PIB between groups (perhaps add into the current figure 3).
6. It will be helpful for the reader if the author can add a flowchart or summary of the processing pipeline.

Minor points

1. Please write 11 in 11C as superscript.
2. Please write the full name for abbreviations and use the abbreviations afterwards in the text. For example BOLD on page 3. SUVR; BPnd on page 4, RT on page 5, ROI on page 11 lack explanation. posterior cingulate cortex at the end of page 2 should be PCC. on page 2, DMN is abbreviation for episodic memory network. The full name of fMRI and PET have not been provided. Please correct.
3. Figure 3 lack indication of the color, and scale bar.

COMMSBIO-21-3698-T

A triple hotspot of Neuroinflammation, Amyloid binding and compensatory task-related brain activity in Alzheimer's Disease, as revealed by dual PET/fMRI

Corresponding Author: Dr. Miguel Castelo-Branco

Authors: Nádia Canário, Lília Jorge, Ricardo Martins, Isabel Santana, Miguel Castelo-Branco

Dear Editor of Communications Biology

We are happy to know that our work is being considered of potential interest for publication in Communications Biology, and we provide below the answers to the reviewers' concerns.

Reviewer #1: This is a brief report of 19 persons with A+ cognitive impairment (presumed due to AD) versus 19 control cognitively unimpaired people (not elevated amyloid) and the relationship of increase in BOLD signal in posterior cingulate in relation to a) amyloid PET levels (11C PIB) and b) a marker of microglial activation (PK11195). The authors find that there was a strong correlation between BOLD signal in the task in fMRI and microglial activation PET tracer, and no relation to amyloid.

Comment 1: While the authors make an (unjustified) therapeutic speculation about the direction of causality of the association, the association is nonetheless of interest. The authors consider only that increased microglial activation was causal, its possible and indeed plausible that the reverse is true, namely that the increased activation is causing the increased microglial activation. See a discussion in Knopman et al, "Alzheimer

Disease Primer" Nature Reviews 2021 on connectivity and its relationship to AD pathology. **Thus, the observations are quite interesting but the interpretation is questionable, and needs to be balanced by the alternative perspective.**

R: We agree that a direction of causality cannot be inferred based on the significant correlation between microglia PET and BOLD response. We now present the alternative perspective as requested.

Comment 2: The main one was that “beta values” was not defined in text, and it took a good deal of scrambling by a reader to appreciate that that term referred to the activations of fMR – there is a mention of what “higher betas” refers to in the legend of Fig 2 and they were referred to again in the Statistical analysis, but there should have been a simple statement in text in Results that clarified the point.

R: We have clarified the term in the main text, which basically refers to the weights (coefficients) of each statistical predictor .

Comment 3: There should be a Table with demographics and basic disease descriptors of the cases and the comparison group.

R: We agree with the reviewer and now provide a table with demographics, disease descriptors as well as the results from the respective comparisons.

Comment 4: I would have liked to see a Figure depicting PIB vs PK11195, in order to understand the relationship of cases to controls.

R: We have added the aforementioned figure.

Comment 5: One other major deficiency was that the relationship to other regions was not described: there is no presentation of the specificity of the findings for post cingulate. Was the BOLD signal – PK11195 signal association present elsewhere? Were there associations with PIB and BOLD elsewhere? These could be included as supplemental files but the reader needs to know.

R: We have done these analyses which confirm effect specificity.

Reviewer #2: The current study investigated the association between task-fMRI activation, microglia PET and amyloid PET across AD patients and cognitively normal controls. Results showed that fMRI activity and microglia PET were positively associated in the PCC.

Overall is the topic of interest. But several concerns compromise the conclusiveness of the results.

Comment 1: The rationale for focusing only on the group contrast for scrambled stimuli in all subsequent analyses and discarding the remainder of the fMRI data is unclear. Also it seems that the voxel-wise group comparison of fMRI activation between AD vs controls just happened to be significant in the PCC and was therefore used for further analysis, whereas the introduction renders the impression that the PCC was selected a priori as the region of interest. Please clarify.

R: Please note that there is a very solid biological justification to choose the PPC, based on imaging and neuropathological data. The choice of PPC was not based on

fMRI data, but on its importance in the pathophysiology of AD. We use several references justifying this point. In any case the scrambled stimuli are the most task demanding, leading to significant behavioral effects, justifying its choice which is further corroborated by task relevant activations.

Comment 2: Were there no brain regions of reduced brain activations in AD compared to controls? This seems surprising. Could the authors provide an explanation for this finding and was that consistent for other stimuli as well?

R: We thank the reviewer for pointing this out and we have indeed found deactivation in the precuneus, which is a region often involved in AD. We now add a comprehensive account of the regions identified in the BOLD map.

Comment 3: Why was a multivariate group comparison done rather than analysing each modality separately? The group differences should be mapped for each modality, so that it is clear in which brain regions increases/decreases in AD were observed.

R: We performed a multivariate group comparison since we wanted to explore the three modalities together and their association with the group. Nonetheless, subsequent post hoc analyses for each modality were also applied. Nevertheless, a univariate analysis for each modality, with further correction for multiple comparisons, would also be adequate.

Comment 4: - In Figure 3 it is hard to discern whether the controls showed abnormally high regional PiB as well. The group comparison would be informative

R: We have now provide group comparisons and have also improved figure 3.

Comment 5: In the methods it is stated that CSF and PIB values were used in order to define abnormal Abeta values for the AD group, but the criteria were not mentioned.

R: We have added in the methods the criteria used in the CSF and PIB to define abnormal values.

Comment 6: - Were all cross-modality correlational analyses done only for the PCC ROI values?

R: We confirm that all cross-modality correlational analysis were hypothesis driven concerning PCC but other analyses were also performed. See also response to reviewer 1.

Comment 7: The lack of a correlation between PiB PET and microglia PET is surprising and contradicts previous reports such as in Hamelin et al. Brain 2016. Please discuss.

R: We agree that the impact of neuroinflammation in Alzheimer's is still highly debated, therefore some studies have indeed found restricted associations between PiB PET and microglia PET, as in Hamelin et al. 2016. However, we for the first-time delineated regions selectively affected by neuropathological mechanisms of AD through an fMRI task, which might account for the divergent findings. Moreover, microglial activation has been found closer to AB in early MCI, whereas in established AD it has been preferentially linked to tau tangle burden. We now discuss this aspect of the manuscript.

Comment 8: On p. 4 (Discussion) the authors refer to an association between “activated microglia and cognition”, but an association with cognitive performance was not reported. In fact, that could have been an interesting addition.

R: We have done a new correlation analysis between PK 11195 and the d-prime taken from the scrambled material for the PCC ROI. We failed to find an association between the two variables (AD: $r = -0.027$, $p > 0.917$; Controls: $r = 0.283$, $p > 0.255$).

Comment 9: Please discuss caveats of the current study.

R: We have added discussion on the caveats of the study.

Comment 10: The potential influence of the rs6971 SNP on binding affinity of the microglia PET tracer is not reported.

R: We now included this issue in the discussion.

Minor:

1. In order to follow the Results section, basic methodological information should be provided to understand what the results refer to. For example, the authors start the Results section by mentioning that the scrambled stimuli were the most difficult to process, but no information on the fMRI paradigm was provided, which renders this result difficult to interpret.

R: We now add further details on the design and materials of the study.

Reviewer #3:

The multimodal imaging study by Canario et al is well designed and of high-interest given the importance of microgliosis in the development of Alzheimer disease. The author described the association between neuroinflammation (TSPO PET), amyloid (PIB PET) and compensatory task related brain activity (fMRI) in 19 AD and 19 control subjects. The author found that microglia activation in the posterior cingulate cortex was associated with increased brain activity in AD patients and independent of amyloid accumulation. More details in the method and results need to be provided to enable understanding of the finding.

Comment 1: Method: Lack of information regarding patient demographics, e.g. age, APOE, gender, MMSE score, disease duration, education. The author only mentioned that the participants are age, gender and education matched without providing detailed information. Neuroinflammation is dynamic and act differently at different disease stage. Thus this information is essential.

R: We fully agree with the reviewer, we have now provided a better characterization of our AD sample.

Comment 2: Method: the author mentioned that the diagnosis was supported by CSF assay. Please specify which CSF assay, and include the results. Is there any correlation between the CSF results and the imaging readouts?

R: We now included the CSF methodology and respective results. We found no correlation between CSF results from AD patients and beta values taken from PCC (B42: $r = -0.117$, $p > 0.678$; TAU: $r = 0.421$, $p > 0.117$; TAU-P: $r = 0.320$, $p > 0.263$; TAU/AB42: $r = 0.020$, $p > 0.944$; B42/TAU-P: $r = -0.279$, $p > 0.333$)

Comment 3: Results: The PK11195 results are not presented. Please add a figure of PK11195 distribution and comparison of BPnd values between groups.

R: We add the results from PK11195 in figure 3, with the respective quantification in PCC.

Comment 4: Please add the cognitive assessment data (if available) which is important for interpretation of the results.

R: We have now included information regarding the cognitive assessment.

Comment 5: Please provide quantification and comparison of the regional SUVR for PIB between groups (perhaps add into the current figure 3).

R: Done. See figure 3.

Comment 6: It will be helpful for the reader if the author can add a flowchart or summary of the processing pipeline.

R: We have added the flowchart of the processing pipeline as supplementary material.

Minor points

1. Please write 11 in 11C as superscript.

R: Done.

2. Please write the full name for abbreviations and use the abbreviations afterwards in the text. For example BOLD on page 3. SUVR; BPnd on page 4, RT on page 5, ROI on page 11 lack explanation. posterior cingulate cortex at the end of page 2 should be PCC.

on page 2, DMN is abbreviation for episodic memory network. The full name of fMRI and PET have not been provided. Please correct.

R: Done.

3. Figure 3 lack indication of the color, and scale bar.

R: Done

Reviewers' comments:

Reviewer #1 (Remarks to the Author):

The authors have been responsive to the concerns of this reviewer. I would probably include the sample size of Alzheimer dementia patients in the Abstract in order to give those persons scanning the Abstract a sense of small number of cases involved in the claims being made.

Reviewer #2 (Remarks to the Author):

Thank you for the revision.

A few issues remain to be addressed:

Amyloid status: Were all AD patients Abeta positive and all controls Abeta negative (assessed by the global abeta levels)?

Figure 3c suggests that at least 3 controls had precuneus levels of Abeta within the pathological range, and figure 4 suggests a substantial overlap in the PCC amyloid PET levels between groups. Also, were the CSF and amyloid PET status consistent?

The authors speculate that pathologic tau rather than Abeta is associated with TSPO PET. The authors now indicate that they also assessed CSF p-tau181. Could the authors test the association between CSF p-tau and TSPO PET or Brain activity in the PCC?

Please show in which brain areas AD differed from controls in the TSPO PET uptake.

Figure 1: Briefly explain task design in the figure legend to render the figure interpretable.

Figure 3: the label c for is used for two different panels.

Please use throughout the manuscript consistent labeling: For example, CSF A β 1-42 and A1-42 are used, Tau-P/p-tau/p-tau(181)/p-tau181 are all used to refer to the same, the TSPO marker is referred to differently (sometimes simply called neuroinflammation even though that is different from TSPO tracer uptake)

It is appreciated that the analyses focused on the PCC. However, is the association between the BOLD contrast and TSPO PET exclusive to that location or also present for the other clusters of altered BOLD activity?

Reviewer #3 (Remarks to the Author):

The author has addressed all the comments.

Reviewers' comments:

We are very pleased to know that both reviewers 1 and 3 were satisfied with our reply to their comments. We also thank reviewer 2 for the additional comments and we now respond to the remaining concerns.

Reviewer #1 (Remarks to the Author):

The authors have been responsive to the concerns of this reviewer. I would probably include the sample size of Alzheimer dementia patients in the Abstract in order to give those persons scanning the Abstract a sense of small number of cases involved in the claims being made.

Response: We have now included the sample size of both groups in the abstract.

Reviewer #2 (Remarks to the Author):

Thank you for the revision.

A few issues remain to be addressed:

1 - Amyloid status: Were all AD patients Abeta positive and all controls Abeta negative (assessed by the global abeta levels)?

Response: We clarified that 3 controls had Abeta values suggestive of being positive but with no evidence of neurological disease nor cognitive dysfunction.

2 - Figure 3c suggests that at least 3 controls had precuneus levels of Abeta within the pathological range, and figure 4 suggests a substantial overlap in the PCC amyloid PET levels between groups. Also, were the CSF and amyloid PET status consistent?

Response: We now stress that all AD patients with abnormal CSF values had consistent abnormal PIB positive values.

3 - The authors speculate that pathologic tau rather than Abeta is associated with TSPO PET. The authors now indicate that they also assessed CSF p-tau181. Could the authors test the association between CSF p-tau and TSPO PET or Brain activity in the PCC?

Response: We have now tested the requested association and we did not find significant correlations between CSF p-tau and brain activity (beta values) nor between CSF p-tau and TSPO PET SUVR.

4 - Please show in which brain areas AD differed from controls in the TSPO PET uptake.

Response: Post hocs show that there are differences in 4 of the total 8 brain areas taken from the GLM-BOLD map, please see table below (statistical significance <0.05 FDR corrected).

Region	t (35)	p
R superior parietal cortex (BA7)	3.662	< 0.007
L Posterior Cingulate cortex (PCC)	2.814	< 0.032
L ventral posterior cingulate gyrus (BA23)	2.459	< 0.047
L superior parietal cortex (BA7)*	2.370	< 0.047
R superior parietal cortex (BA7)*	2.183	> 0.056
R Fusiform Gyrus (BA37)*	1.784	> 0.111
L Prefrontal Gyrus (BA10)	-0.876	> 0.441
R Insula	0.424	> 0.674

*depicts deactivation ROIs

5 - Figure 1: Briefly explain task design in the figure legend to render the figure interpretable.

Response: We have explained the task design in the figure legend.

6 - Figure 3: the label c for is used for two different panels.

Response: We have now corrected the label in the figure.

Please use throughout the manuscript consistent labelling: For example, CSF A β 1-42 and A1-42 are used, Tau-P/p-tau/p-tau (181)/p-tau181 are all used to refer to the same, the

TSPO marker is referred to differently (sometimes simply called neuroinflammation even though that is different from TSPO tracer uptake)

Response: We agree and now used consistent labelling and explain that we interpreted the TSPO tracer uptake as an index of neuroinflammation.

7 - It is appreciated that the analyses focused on the PCC. However, is the association between the BOLD contrast and TSPO PET exclusive to that location or also present for the other clusters of altered BOLD activity?

Response: We confirm that this association was indeed specific to this region.

Reviewer #3 (Remarks to the Author):

The author has addressed all the comments.